# Deciphering the Genetic Code of Autoimmune Kidney Diseases

**DOI:** 10.3390/genes14051028

**Published:** 2023-04-30

**Authors:** Stephanie U-Shane Huang, Oneli Kulatunge, Kim Maree O’Sullivan

**Affiliations:** Department of Medicine, Centre for Inflammatory Diseases, Monash University, Clayton, VIC 3168, Australia

**Keywords:** gene mutations, inflammation, glomerulonephritis, neutrophil extracellular traps, anti-neutrophil cytoplasmic antibody vasculitis, lupus nephritis, IgA nephropathy anti-glomerular basement disease

## Abstract

Autoimmune kidney diseases occur due to the loss of tolerance to self-antigens, resulting in inflammation and pathological damage to the kidneys. This review focuses on the known genetic associations of the major autoimmune kidney diseases that result in the development of glomerulonephritis: lupus nephritis (LN), anti-neutrophil cytoplasmic associated vasculitis (AAV), anti-glomerular basement disease (also known as Goodpasture’s disease), IgA nephropathy (IgAN), and membranous nephritis (MN). Genetic associations with an increased risk of disease are not only associated with polymorphisms in the human leukocyte antigen (HLA) II region, which governs underlying processes in the development of autoimmunity, but are also associated with genes regulating inflammation, such as *NFkB, IRF4*, and FC γ receptors (*FCGR*). Critical genome-wide association studies are discussed both to reveal similarities in gene polymorphisms between autoimmune kidney diseases and to explicate differential risks in different ethnicities. Lastly, we review the role of neutrophil extracellular traps, critical inducers of inflammation in LN, AAV, and anti-GBM disease, where inefficient clearance due to polymorphisms in DNase I and genes that regulate neutrophil extracellular trap production are associated with autoimmune kidney diseases.

## 1. Introduction

Autoimmunity arises due to the loss of tolerance to self-antigens, leading to the activation of autoreactive immune cells that cause damage to major organs, such as the kidneys [1,2]. Kidney nephrons act as a high-pressure filtration system composed of a glomerulus and a tubule. The glomerulus, which consists of specialised membranes to filter blood, is vulnerable to the deposition of autoantibodies or the infiltration of leukocytes [2,3]. Activated autoreactive cells can subsequently secrete inflammatory mediators and cause the further accumulation of cells, directly interacting with renal cells to cause glomerulonephritis (GN) and tissue damage [4,5,6]. If left untreated, patients can develop end-stage kidney disease (ESKD), where their survival would be dependent on dialysis or a kidney transplant [7].

Kidney disorders can occur as a consequence of systemic autoimmunity, as in the case of systemic lupus erythematosus (SLE) and anti-neutrophil cytoplasmic antibody (ANCA)-associated vasculitis (AAV). SLE is a chronic inflammatory disease characterised by the production of autoantibodies against nuclear DNA. In 35–55% of patients, abnormal inflammation in the glomeruli can occur, correlating to the development of lupus nephritis (LN) [8,9]. AAV on the other hand, is a rare autoimmune disease of small and medium blood vessels, caused by the loss of tolerance to neutrophil granule proteins proteinase 3 (PR3) or myeloperoxidase (MPO), and is characterised by the presence of anti-PR3 or anti-MPO autoantibodies, respectively [4,5]. Furthermore, autoimmune kidney diseases such as IgA nephropathy (IgAN), the most prevalent type of GN, can be mediated through deposition on nonspecific IgA autoantibodies and immune complexes in the glomeruli [3,10,11,12]. Specific cell types in the glomerulus can provide a source of autoantigens, resulting in membranous nephropathy (MN,) or glomerular basement membrane (GBM) nephropathy (Goodpasture’s disease) [3]. 

As with other diseases, causes of autoimmune kidney diseases are multifactorial, with numerous environmental and genetic factors involved. Polygenic autoimmunity is a common occurrence; however, in rare cases, monogenic autoimmune diseases caused by single gene mutations can occur [13]. Approximately 90% of genetic variants associated with autoimmune kidney diseases are encoded in non-coding regions, or introns, that are involved in epigenetic regulation (i.e., turning genes on or off and regulating their activity), and roughly 60% lie within immune-cell enhancers [14]. 

As several reviews have covered genetic associations of the autoimmune kidney diseases discussed above, the focus of this review will be on recently characterised polymorphisms, the relevance of single-nucleotide polymorphisms (SNPs) that influence effector cells in disease pathogenesis, and neutrophil extracellular traps (NETs), critical inducers of inflammation in glomerulonephritis (in LN, AAV, and anti-GBM disease). 

## 2. Known Gene Associations of Autoimmune Kidney Diseases

### 2.1. Lupus Nephritis

LN is an autoimmune disease that disproportionately affects women (nine out of 10 are women) and generally strikes in their child-bearing years. Asians and African Americans are more likely to develop LN compared to Caucasians. Genetic association studies have identified over fifty SNPs that may underlie disease pathogenesis in LN [15]. These genetic variants include pathways associated with altered programmed cell death (PCD) and the dysfunctional immune clearance of PCD debris, which may contribute to the generation of immune complexes that contain autoantibodies. A range of polymorphisms affect each phase of immunity, from the initial innate immunity phase to the amplifying and adaptive phases of the immune response. Lastly, genetic variants influence the kidney-specific effector response, which can impact organ damage and progression to ESRD and mortality. While these genetic polymorphisms have been covered extensively in these reviews [16,17] and summarised in Table 1 and Figure 1, this article will discuss specific SNPs that impact immune effector functions (summarised in Table 2) and recently identified genetic variants.

**Table 1 genes-14-01028-t001:** Heatmap showing genetic polymorphisms associated with autoimmune kidney diseases. The chromosome locations and identification numbers (ID) for single-nucleotide polymorphisms are shown for each gene found to have associations with lupus nephritis (LN), myeloperoxidase anti-neutrophil cytoplasmic antibody-associated vasculitis (MPO-AAV), proteinase 3 AAV (PR3-AAV), IgA nephropathy (IgAN), or anti-glomerular basement membrane disease (anti-GBM).

Chromosome	Gene Name	SNP ID	LN	MPO−AAV	PR3−AAV	IgAN	Anti−GBM	References
6p21.32	*HLA−DP*	rs3117242			6.20 × 10^−5^			[18,19,20]
6p21.32	*HLA−DP*	rs9277535				1.00 × 10^−03^		[10,21]
6p21.32	*HLA−DPA1*	rs9277341			4.52 × 10^−84^			[18,22]
6p21.32	*HLA−DPB1*	rs141530233			1.33 × 10 ^−106^			[18,22]
6p21.32	*HLA−DPB1*	rs1042169			6.53 × 10^−106^			[18,22]
6p21.32	*HLA−DR3*	rs2187668	3.70 × 10^−05^					[17,23,24]
6p21.32	*HLA−DR4*		1.00 × 10^−03^					[16,25]
6p21.32	*HLA−DR11*		1.00 × 10^−04^					[16,25]
6p21.32	*HLA−DR15*		1.00 × 10^−03^					[16,17,25]
6p21.32	*HLA−DRB1*1501*	rs3135388					1.00 × 10^−04^	[26]
6p21.32	*HLA−DQA1*	rs35242582			5.78 × 10^−18^			[18,22]
6p21.32	*HLA−DQA2*	rs3998159		5.24 × 10^−25^				[18,22]
6p21.32	*HLA−DQA2*	rs7454108		5.03 × 10^−25^				[18,22]
6p21.32	*HLA−DQB*	0602					1.00 × 10^−04^	[27]
6p21.32	*HLA−DQB*	0501					1.00 × 10^−03^	[27]
6p21.32	*HLA−DQB1*	rs1049072		2.13 × 10^−24^				[18,22]
14q32.13	*SERPINA1*	rs28929474			3.09 × 10^−12^			[18,22]
14q32.13	*SERPINA1*	rs7151526			5.61 × 10^−12^			[18,19]
6p21.32	*R × RB*	rs6531			5.21 × 10^−05^			[28,29]
1p13.2	*PTPN22*	rs2476601		4.11 × 10^−06^	6.10 × 10^−09^			[18,30,31]
1p13.2	*PTPN22*	rs6679677		1.88 × 10^−08^			[18,22]
2q33.2	*CTLA4*	rs3087243		6.4 × 10^−03^			[20,22,30]
2q33.2	*CTLA4*	rs231777				3.60 × 10^−02^		[10,32]
19p13.3	*PRTN3*	rs62132295(A)			2.61 × 10^−07^			[33]
19p13.3	*PRTN3*	rs62132293			3.59 × 10^−13^			[33]
3p21.2	*TLR9*	rs352140			1.10 × 10^−05^			[34]
3p21.2	*TLR9*	rs352162			3.98 × 10^−06^			[34]
1q23	*FCGR2B*	rs1050501					2.80 × 10^−04^	[35]
1q23	*FCGR3A*	rs396991	1.75 × 10^−02^					[17,36,37]
1q23	*FCGR3B*	CNV	4.00 × 10^−03^	2.90 × 10^−04^	3.00 × 10^−03^			[20,38,39]
1q23	*FCGR3B*	CNV			1.10 × 10^−04^			[20,39]
1q23	*FCGR2A*	CNV					1.30 × 10^−02^	[40]
1q23	*FCGR2A*	rs2205960	1.30 × 10^−03^					[16,17,41,42]
1q23	*FCGR2A*	rs1048265	5.00 × 10^−03^					[42]
1q32.1	*IL10*	rs1554286		1.00 × 10^−06^				[19,43]
10p15.1	*IL2RA*	rs41295061		1.22 × 10^−02^			[19,44]
7p22.1	*C1GALT1*	rs5882115				4.00 × 10^−03^		[10,45]
7q36.1	*NOS3*	rs1799983				2.90 × 10^−02^		[10,46]
2q33.2	*ICOS*	rs4270326				4.10 × 10^−02^		[10,32]
2q33.2	*ICOS*	rs4404254				4.30 × 10^−02^		[10,32]
12q15	*IFNG*	rs430561				4.30 × 10^−02^		[10,47]
12q15	*IFNG*	microsatellite 114	4.60 × 10^−02^					[16,48]
18p11.31	*TGFB1*	rs1800469				8.00 × 10^−03^		[10,49]
18p11.31	*TGFB1*	rs6957				1.70 × 10^−02^		[10,49]
19q13.33	*KLK1*	rs5516	8.00 × 10^−03^					[50]
19q13.33	*KLK1*	rs2740502	1.00 × 10^−02^					[50]
16p11.2	*ITGAM*	rs1143679	2.22 × 10^−21^					[17,51]
4q24	*BANK1*	rs4699261	3.30 × 10^−07^					[52]
2q32.2	*STAT4*	rs7582694	1.50 × 10−^02^					[53]
2q32.2	*STAT4*	rs11880341	3.70 × 10^−09^					[24]
19p13.2	*TYK2*	rs2304256	2.60 × 10^−02^					[54,55]
19p13.2	*TNIP1*	rs7708392	1.82 × 10^−11^					[17,24,56,57]
19p13.2	*TNIP1*	rs4958881	4.43 × 10−^03^					[17,24,56,57]
1p13.3	*IL23R*	rs7517847					4.60 × 10^−02^	[58]
6p12.2	*IL17A*	rs2275913					7.00 × 10^−03^	[58]
16p13.3	*DNASEI*	rs1053874	5.00 × 10^−02^					[59,60,61]
3p14.3	*DNASEIL3*	rs35677470	4.20 × 10^−7^					[62]
3p21.31	*TRE × 1*	rs56203834	2.99 × 10^−13^					[63]

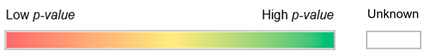

**Table 2 genes-14-01028-t002:** Functionality of proteins translated from genes associated with autoimmune kidney diseases.

Chromosome	Gene Name	Gene Function	References
6p21.32	*HLA-DP*	One of 3 loci (together with HLA-DQ and HLA-DR) responsible for encoding the Major Histocompatibility Complex (MHC) class II molecule.	[20,64,65]
6p21.32	*HLA-DPA1*	HLA class II transmembrane heterodimer expressed on antigen-presenting cells involved in the presentation of extracellular proteins.	[66]
6p21.32	*HLA-DPB1*	One of the molecules in the HLA class II heterodimer, encoding the β chain.	[66]
6p21.32	*HLA-DR3*	An allele of HLA-DR and the most established variant for SLE susceptibility.	[17,67]
6p21.32	*HLA-DR4*	An allele of HLA-DR proposed to be a protective factor for the development of SLE and LN.	[16,25]
6p21.32	*HLA-DR11*	An allele of HLA-DR proposed to be a protective factor for the development of SLE and LN.	[16,25]
6p21.32	*HLA-DR15*	An allele of HLA-DR proposed to be a high risk factor for the development of SLE and LN.	[16,17,25]
6p21.32	*HLA-DRB1*1501*	An allele of the DRB1 gene, belonging to the HLA class II β chain paralogues, associated with susceptibility to anti-GBM disease.	[26,27,66]
6p21.32	*HLA-DQA1* and *HLA-DQA2*	Belongs to the α chain of HLA class II molecules.	[22,66]
6p21.32	*HLA-DQB* and *HLA-DQB1*	A heterodimer belonging to the HLA class II β chain paralogues.	[27,66]
14q32.13	*SERPINA1*	Serpin family A member 1 (SERPINA1) encodes α-1 antitrypsin, the primary inhibitor of neutrophil granule protein, PR3.	[18,19,20,22]
6p21.32	*RXRB*	Retinoid X receptor B (RXRB) encodes a family of retinoid X receptors that function as transcription factors involved in development, cell differentiation, metabolism, and cell death.	[28,29,68]
1p13.2	*PTPN22*	Encodes protein tyrosine phosphatase non-receptor type 22 (PTPN22), which negatively regulates signalling downstream of TLRs, T cell receptors, and B cell receptors via the dephosphorylation of intermediate molecules including the Src and Syk family kinases.	[22,30,69,70]
2q33.2	*CTLA4*	Cytotoxic T-lymphocyte-associated protein 4 (CTLA4) is involved in the regulation of homeostasis via the inhibition of activated T cell responses. CTLA4 has a higher affinity for CD80/86 (found on antigen-presenting cells) and outcompetes co-stimulatory protein CD28.	[20]
19p13.3	*PRTN3*	Proteinase 3 (PRTN3) encodes the primary neutrophil granule protein PR3.	[33]
3p21.2	*TLR9*	Toll-like receptor 9 (TLR9) encodes the endosomal pattern recognition receptor TLR9, which recognises DNA, particularly unmethylated DNA (CpG motifs), and activates the NF-kB signalling pathway.	[34,71]
1q23	*FCGR2A*	Fc γ receptor 2a (FCGR2A) is a low-affinity receptor expressed on multiple cell types to mediate phagocytosis, release of inflammatory cytokines, and clearance of immune complexes.	[72]
1q23	*FCGR2B*	Fc γ receptor 2b (FCGR2B) is a low-affinity receptor involved in phagocytosis of immune complexes and regulates B cell antibody production.	[66]
1q23	*FCGR3A*	Fc γ receptor 3a (FCGR3A) is a receptor involved in antibody-dependent natural killer cell cytotoxic activity.	[73]
1q23	*FCGR3B*	Fc γ receptor 3b (FCGR3B) is a low-affinity receptor expressed by neutrophils and eosinophils. It is also involved in the tethering and clearance of immune complexes by neutrophils.	[74]
1q32.1	*IL10*	Interleukin 10 (IL10) is a potent anti-inflammatory cytokine that dampens the host immune response to pathogens, preventing damage and maintaining tissue homeostasis.	[75]
10p15.1	*IL2RA*	Interleukin 2 receptor α (IL2RA) is expressed on multiple leukocytes, with highest expression found on regulatory T cells. Receptor activation promotes cell survival, proliferation, and differentiation.	[76,77]
7p22.1	*C1GALT1*	Encodes a glycosyltransferase that is involved in multiple vital biological functions such as angiogenesis, platelet production, and kidney development.	[78,79]
7q36.1	*NOS3*	Nitric oxide synthase 3 (NOS3) is an enzyme that synthesises nitric oxide, a bioactive nitrogen molecule with physiological roles in kidney, cardiovascular, and metabolic systems.	[80]
2q33.2	*ICOS*	Inducible T cell co-stimulator (ICOS) is a co-stimulatory molecule expressed on activated CD4 and CD8 T cells; it promotes the induction, maintenance, and homing of T follicular helper cells.	[81,82]
12q15	*IFNG*	Interferon γ (IFNg) is a major effector cytokine in immune defence, primarily secreted by CD4 T helper 1 (Th1) cells and CD8 cytotoxic T cells. It can exert anti-tumour effects and suppress autoimmune processes.	[83]
18p11.31	*TGFB1*	Encodes the cytokine transforming growth factor β 1 (TGFB1), responsible for mediating cell proliferation, differentiation, IgA isotype switching, and the upregulation of IgA synthesis. TGFB1 also has fibrotic properties to which the kidneys are particularly sensitive.	[49,84]
19q13.33	*KLK1*	Encodes the serine protease kallikrein (KLK1) and is involved in anti-inflammatory, anti-apoptotic, anti-fibrotic, and anti-oxidative roles.	[50,85]
16p11.2	*ITGAM*	Encodes CD11b-integrin (α M chain), a subunit of complement receptor 3 (CR3), which is expressed widely on innate immune cells and allows for the activation of the complement cascade.	[17,51]
4q24	*BANK1*	B cell scaffold protein with ankyrin repeats (*BANK1*) is expressed on B cells and is involved in the phosphorylation of phospholipase C (PLC) and phosphoinositide 3-kinase (PI3K), both of which are important in B cell signalling pathways.	[52,86]
2q32.2	*STAT4*	Encodes for the signal transducer and activator of transcription factor 4 (STAT4), primarily activated by IL-12, and is essential in the promotion of Th1-mediated immune responses. STAT4 also has a role in the activation of the transcription of IFNy.	[24,53,87]
19p13.2	*TYK2*	Encodes for the enzyme tyrosine kinase 2 (TYK2), a member of the Janus kinase (JAK) family, involved in the signal transduction of type I IFNs (IFNα/β), IL-6, IL-10, IL-12, and IL-23.	[54,55,88]
19p13.2	*TNIP1*	TNFAIP3 Interacting Protein 1 (TNIP1) encodes the adaptor protein ABIN1, involved in the regulation of NF-kB activation and cell death signalling.	[17,66,89]
1p13.3	*Il23R*	Receptor for interleukin 23 (IL23), associates constitutively with JAK2, and binds to STAT3 upon ligand binding.	[66]
6p12.2	*IL17A*	Interleukin 17 a (IL-17A) is a cytokine produced by activated T cells that mobilises and activates neutrophils.	[90]
16p13.3	*DNASE1*	DNase I preferentially degrades dsDNA; primary role is to degrade extracellular nuclear proteins.	[91]
3p.14.3	*DNASE1L3*	Deoxyribonuclease I Like 3 encodes for DNase γ; primary role in intra- and extracellular degradation of DNA.	[91]
3p21.31	*TREX1*	Also known as *DNASEIII*, it is a cytosolic DNA nuclease, digests single-stranded DNA, and is critical for the regulation of cGAS-STING signalling.	[63]
19q13.1	*NSPH1*	Encodes for nephrin, a protein found in podocytes, and is required for proper functioning of the renal filtration barrier.	[92]
1q25-31	*NSPH2*	Encodes for podocin, a protein found in podocytes, and is required for proper functioning of the renal filtration barrier.	[93]
1q31.3	*CFH*	Encodes for a complement control protein that regulates the alternative pathway of the complement system.	[94]

**Figure 1 genes-14-01028-f001:**
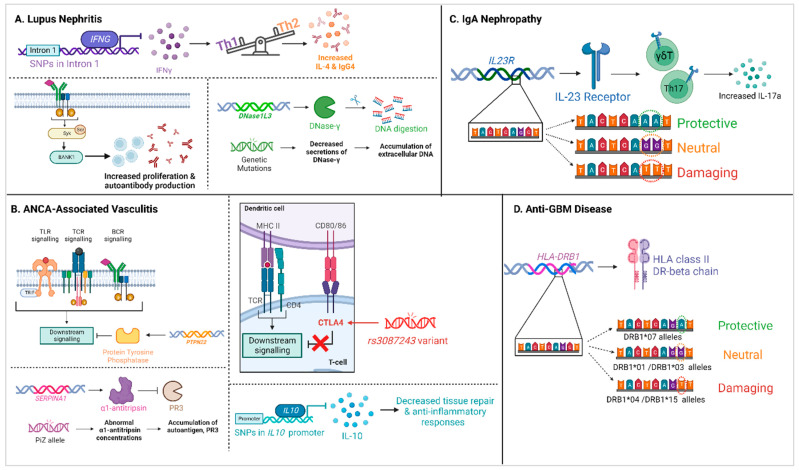
Summary of genetic variants affecting autoimmune kidney diseases. (**A**) depicts the SNPs affecting lupus nephritis, particularly variants in the *IFNG, BANK1,* and *DNase1L3* genes [48,52,95,96,97,98,99,100,101]. (**B**) depicts how genetic variations in *PTPN22, CTLA4, SERPINA1,* and *IL10* affect the development of ANCA-associated vasculitis [20,30,43,69,102,103]. (**C, D**) depict the hierarchy of genetic variants of *IL23R* and *HLA-DRB*, respectively, and its effects on (**C**) IgA nephropathy and (**D**) anti-GBM disease [11,26,27,58,104,105,106,107,108]. Figure created using BioRender.com (15 November 2022).

Polymorphisms in the major histocompatibility complex (MHC) region were the first to be characterised as LN risk factors. The MHC region contains the human leukocyte antigen (HLA) genes, many of which encode receptors, cytokines, and transcription factors that underlie immune function [109]. HLA-DR15 and HLA-DR3 were found to be risk alleles for LN, whereas HLA-DR4 and HLA-DR11 were associated with protection from disease [25]. Although strong associations between these genes and LN have been shown, no mechanistic studies have demonstrated their direct relationship with disease pathogenesis. One study showed that a dominant HLA-DR3 restricted T cell epitope on the SmD autoantigen and its mimicry peptides could activate T cells that recognised this epitope [110]. Another study demonstrated the association between HLA-DR3 and anti-Ro autoantibodies in SLE patients [111]. These studies suggest that the HLA-DR3 polymorphism may cause structural changes in the epitope that enhance T cell activation, thereby triggering the activation of autoreactive B cells, resulting in the production of autoantibodies.

Polymorphisms that affect the interferon signalling pathway are also associated with LN, such as the interferon γ *(IFNG*) gene and signal transducer and activator of transcription 4 (STAT4). The relationship between *STAT4* and LN was demonstrated in a Vietnamese cohort, where a significant difference of *p* = 0.015 was observed in the distribution of the *STAT4* genotype at position rs7582694 in SLE patients with LN compared to controls [53]. One study showed a stronger association in SNP rs11889341 with LN amongst a Swedish cohort, with *p* = 3.7 × 10^−9^ [24]. STAT4 is expressed in T cells and is phosphorylated in response to interleukin 2 (IL2), leading to the transcriptional activation and production of IFNγ [112,113,114]. This is a critical process that drives the maturation of T helper 1 (Th1) cells [114,115], which have been shown to be important effectors in LN pathogenesis [116], where polymorphisms in *STAT4* could underlie a functional change that enhances Th1 activation, accelerating disease progression.

The genetic association of *IFNG*¸ the cytokine secreted by Th1 cells, has been demonstrated with in vitro evidence. Miyake and colleagues documented ten alleles within the first intron of *IFNG*, starting at microsatellite position +875, ranging from 106 to 124 base pairs (bp) in length [48]. These alleles contained repeating dinucleotide sequences, with each being designated according to the number of amplified DNA bps. *IFNG* 114, with thirteen CA repeats, was found to be significantly elevated in patients with LN compared to healthy controls. Healthy controls with the *IFNG* 114 genotype were found to have blood mononuclear cells (PBMC) that secreted less IFNγ, and a lower ratio of IFNγ producing CD4^+^ T cells compared to individuals without the polymorphism. As IFNγ is secreted by Th1 cells and required for their maturation, reduced production of IFNγ may inhibit the development of Th1 cells and skew the microenvironment toward a Th2 phenotype. These cells produce interleukin 4 (IL-4) and IgG4, which can deposit in the microvasculature and basement membrane of the glomerulus, leading to the development of membranous GN (shown in Figure 1). 

B Cell Scaffold Protein with Ankyrin Repeats 1 (*BANK1*) was also recently shown to be strongly associated with the severe form of proliferative LN [52]. BANK1 is a B cell adaptor protein that is functionally related to toll-like receptor (TLR) pathways that activate Fib and IFN regulatory factors that enhance B cell activation, inflammation, and characteristic features of SLE [95,96,97,98]. The nonspecific activation of autoreactive B cells produces the hallmark polyclonal autoreactive antibodies in LN [117]. Furthermore, B cells can activate autoreactive T cells and produce local proinflammatory effects [99]. The SNP may enhance B cell activation, promoting its proinflammatory effector functions and accelerating the pathogenesis of LN.

Common variants in Deoxyribonuclease 1 Like 3 (*DNASE1L3*) have been associated with a slight elevation in the risk of SLE; the GWAS-linked SNP in *DNASE1L3* associated with these diseases does not completely inhibit the function of DNase, but decreases its levels of secretion [59]. *DNASE1L3* encodes the DNase-γ, which digests the DNA of dead cells. Its absence leads to the accumulation of extracellular DNA that stimulates an immune response that can trigger an autoimmune response, where antibodies to DNA and chromatin are produced [101]. While polymorphisms in *DNASE1L3* have been associated with SLE, it has not been associated with the risk of LN. However, a recent study showed that a patient with LN possessing a homozygous variant in *DNASE1L3* at position rs74350392 had the lowest level of DNase activity, with high remnants of NET components in the serum [118]. As DNase acts to degrade DNA, and NETs are DNA structures, a reduction in DNase activity in the LN patient therefore prevented the digestion of NETs, exacerbating the development of renal injury (to be discussed in more detail in a further section). It is important to note that low activity of these genes results in low expression of DNase I or DNase 1L3 in the serum, where a burden of extracellular DNA from the death of dying cells is ineffectively removed, contributing to inflammation. The two genes should be considered together as it is possible that in the absence of one, the other gene and a subsequent endonuclease compensate. SNPs in either *DNAS*E1 or *DNASE1L3* can result in low activity of endonucleases and may be a critical factor in determining susceptibility to not only autoimmune kidney disease but other autoimmune associations as well [91]. 

### 2.2. MPO and PR3 ANCA-Associated Vasculitis

AAVis a relatively rare autoimmune disease that, if left untreated, can advance to rapidly progressive GN (RPGN). AAV is caused by the development ANCA to either the neutrophil enzymes MPO or PR3, which activate neutrophils to release injurious enzymes through degranulation, or the release of NETs that damage bystander tissue. AAV affects men and women equally and manifests on average at around 50-60 years of age.

As genetic variants associated with MPO-AAV (microscopic polyangiitis, MPA) and PR3-AAV (granulomatosis with polyangiitis, GPA) vasculitis have been extensively discussed in recent reviews [33,119] and summarised in Table 1 and Figure 1, this current article will only briefly touch on selected SNPs in AAV disease aetiology. 

A systematic review and analysis of GWAS studies revealed 20 SNPs associated with MPO-ANCA and 134 with PR3-ANCA vasculitis, with no overlapping SNPs [22]. GO enrichment analysis was performed to understand which main gene sets these SNPs belonged to, showing that the processing of antigens via MHCII, the IFNγ-mediated pathway, and the T cell receptor signalling pathway were the main biological processes disrupted in both MPO-AAV and PR3-AAV. Most genetic variants were detected in the HLA region, where polymorphisms result in amino acid changes in MHCII, causing structural changes that influence peptide-binding properties and antigen presentation [120]. However, there is also compelling evidence that implicates the involvement of various non-HLA genes in the development and exacerbation of disease [22,28,29,30,69,102,103,121,122].

#### 2.2.1. HLA Region SNPs

The European Vasculitis Genetics Consortium performed the first GWAS study in AAV that demonstrated distinct genetic differences between GPA and MPA, where *HLA-DPB1* and *HLA-DPA1* were associated with GPA, and *HLA-DQB1* and *HLA-DQA2* with MPA [19]. HLA-DPB genes belong to the HLA class II β chain paralogues, where the class II molecule consists of α (DPA) and β (DPB) chains that form a heterodimer; this complex is expressed on antigen-presenting cells and functions to present peptides. Polymorphisms can occur within the α or β chain, which can alter the peptide-binding specificity, which is associated with a higher risk of GPA. Similar to HLA-DPB, HLA-DQB genes, which were found to be associated with MPA and also belong to the HLA class II β chain paralogues, these heterodimers consist of DQA α and DQB β chains instead of DPA and DPB chains, and polymorphisms within these chains also affect the binding affinity [66]. Differences in the amino acid sequences of HLA proteins may affect the ways in which they bind and present peptides, potentially leading to differences in the types of immune responses that they stimulate. HLA alleles can bind a wide range of peptides, and different alleles may have different peptide-binding preferences. Studies showed that these polymorphisms had strong associations with ANCA specificity, with others showing associations with disease severity, prognosis, and relapse [123,124]. As AAV is an autoimmune disease, the identification of polymorphisms in the HLA region that determine processes that underlie the development of autoreactivity provides insight into the disease pathogenesis of the distinct AAV subtypes, creating unique opportunities for therapeutic strategies.

#### 2.2.2. Non-HLA Region SNPs

Most of the non-HLA region SNPs encode membrane proteins, where the identified polymorphisms alter the signalling pathways for immune cell function, resulting in a higher risk of disease, or provide protection against disease manifestation. One such membrane protein is cytotoxic T lymphocyte-associated antigen 4 (*CTLA-4*); it encodes an inhibitory surface protein on activated T cells that acts as a competitive ligand for CD80 and CD86 [102]. rs3087243 in *CTLA4* has been associated with AAV using both genotype and allele tests [30], where the minor allele of this SNP enhanced protection.

Receptors with SNPS include: *TLR9*, Fc γ receptor (FCGR), and IL-2 receptor subunit α (IL2RA). TLR9 encodes an innate recognition receptor that responds to unmethylated CpG motifs present in bacterial DNA [125]. A publication demonstrated the contribution of TLR9 polymorphisms to PR3-AAV susceptibility and clinical presentation [34]. While *Staphylococcus aureus (S. aureus)* was believed to be a potential trigger for AAV traditionally [126,127], recent publications show that these studies were flawed and do not accurately represent a solid association between *S. aureus* infection and disease pathogenesis [128,129]. Perhaps a dysfunctional response to infection may be the determinant of AAV susceptibility. Further investigation is required to elucidate the mechanisms underlying the correlation between TLR9 variants and AAV aetiology.

FCGRs are expressed on numerous cell types with varying affinities for the Fc regions of IgG subclasses, with the function of regulating inflammation and immune responses [130]. Strong associations were shown for both MPO-AAV and PR3-AAV with the *FCGR3B* copy number, where differences in the copy numbers were associated with disease [39,131]. A low copy number of *FCGR2B* was found to be associated with systemic inflammation, which commonly affected the glomerulus [39]. As *FCGR3B* is responsible for the tethering of neutrophils and eosinophils to immune complexes and subsequent clearance [74], individuals with low copy numbers of the gene may have reduced clearance of immune complexes, leading to their deposition in the glomerular microvasculature, triggering an inflammatory response, which, if chronic, may develop into autoimmunity. Moreover, enhanced neutrophil activation by ANCA may be associated with reduced or absent expression of *FCGR3B* [39], providing further evidence for the involvement of altered copy numbers in the pathogenesis of ANCA vasculitis.

*IL2RA* encodes a high-affinity receptor on the surfaces of T cells, activated B cells, natural killer cells, and monocytes, to modulate immune function [132]. The binding of IL-2 to IL2RA controls survival, proliferation, T cell activation, and normal regulatory T cell (Treg) development [133]. An association was found between an SNP at position rs41295061 and AAV, suggesting that dysfunctional immune signalling through IL2RA may be responsible for disease pathogenesis. 

The last AAV-associated receptor gene to be discussed in this review is protein tyrosine phosphatase, non-receptor type 2 (PTPN22), which encodes protein tyrosine phosphatase in lymphoid cells and is responsible for negatively regulating signalling downstream of TLRs, T cell receptors, and B cell receptors [69], where the variant rs2476601 has been shown to result in abnormal Treg activity and increased neutrophil function [122]. The SNP rs2476601 has also been associated with AAV, with the minor allele conferring disease susceptibility [30].

As cytokines are crucial in the regulation of the immune response [134], it was anticipated that polymorphisms in genes encoding these molecules would be associated with autoimmunity. The *IL10* gene encodes an anti-inflammatory cytokine, interleukin 10, secreted by Th2 cells [135,136], where enhanced expression has been detected in patients with AAV [137]. SNPs in the IL-10 gene promoter (IL-10-3575, IL-10-1082, and IL-10-592) are responsible for producing the anti-inflammatory cytokine IL-10 and have been associated with the development of GPA and ANCA- EGPA, both of which are varying categories of AAV [43].

### 2.3. IgA Nephropathy

IgAN, also known as Berger’s Disease, is caused by the deposition of IgA antibodies within the kidney glomeruli. It is one of the most common autoimmune kidney diseases and can occur at any age. Worldwide, IgA more commonly affects Asian people than Caucasians. In the United States of America, IgAN is more common in males than females and presents in the late teens to early thirties. There is evidence that IgAN can be associated with a family history of either IgAN or IgA vasculitis [10]. 

While polymorphisms associated with IgAN have been covered extensively in a recent review [10] and preprint [138], this current review has summarised the SNPs in Table 1. The referenced publications found that the genetic variants influenced the immunological and inflammatory pathogenesis of IgAN, where proteins of the recently identified genes interleukin 23 receptor (IL23R) and IL17A are also implicated in disease pathology. 

IL23R and IL-17A were recently associated with IgAN susceptibility for the first time in a Chinese Han population [58]. A comparative analysis between genotype distributions, clinical indexes, and pathological grades was carried out between 164 IgAN patients and 192 healthy controls. The analysis revealed that the GG genotype in rs7517847 was associated with a lowered IgAN risk compared to the TT genotype. Another polymorphism in rs2275913 was detected, where the AA genotype was related to a decreased IgAN risk compared to the GG genotype. IL23R subunits were shown to be present on γδ T cells and Th17 cells, which produce IL-17A [104]. γδ T cells have been found to infiltrate the kidneys in patients with progressive IgAN [105] and the imbalance of Tregs and Th17 cells was also found in IgAN patients [106].

CTLA-4 polymorphisms are associated with numerous nephrotic disorders, including minimal change disease (MCD), membranous GN, and focal segmental glomerulosclerosis (FSGC). The SNP (+49A/G) correlates with disease severity. A decrease in CTLA-4 expression has been attributed to polymorphisms in +49A/G and CT60 G/A, and it has been suggested that this may be an independent risk factor for increased proteinuria in IgAN patients [139].

### 2.4. Anti-Glomerular Basement Membrane Disease

Anti-GBM disease, or Goodpasture’s disease, is characterised by autoantibodies to the NC1 domain of the α3 chain of type IV collagen within the basement membrane of both the kidney glomeruli and lungs [27]. It is a rare autoimmune disease that is fatal if not treated immediately as it results in a rapid decline in kidney function and ESKD in a relatively short time. It is classified as a small-vessel vasculitis, resulting in glomerular crescent formation in the kidney and alveolar haemorrhage in the alveoli of the lungs. It is well characterised in both Asian and Caucasian populations but is less common in individuals with African ancestry. Strong associations with HLA-DRB1 genes have been characterised [27]. It is not known what causes the loss of tolerance to type IV collagen, but there are anecdotal reports of associations with seasonal infections, particularly with influenza A. Environmental factors, particularly inhaled hydrocarbons, have the potential to induce inflammation and structurally alter the glomerular and alveolar membranes, allowing antigenic exposure for autoantibodies [140]. It is plausible that both environmental factors and infectious agents trigger disease onset due to genetic susceptibilities. A hierarchy of genetic variations at the DRB1 locus have been identified to confer differential associations with anti-GBM disease. DRB1*15 alleles were identified to have greater susceptibility to disease over DRB1*04; DRB1*03 alleles were neutral in disease aetiology; DRB1*01 alleles were weakly protective; and DRB1*07 alleles were found to be protective against anti-GBM [26,27]. Goodpasture’s autoantibodies bind two epitopes of the α3 type IV collagen chain [107]; the protective HLA-DR molecules from DRB1*01 and DRB1*07 bind the human T cell epitopes with higher affinity, preventing their presentation to DRB1*15 [108], thereby maintaining tolerance.

The *FCGR* and *KLK* gene families have also been associated with anti-GBM disease [141]. Fc γ receptors (FCγR) have been shown to play an important role in the protection against anti-GBM disease pathogenesis in mouse models [142,143,144]. As opposed to other autoimmune kidney diseases such as MPA, GPA, and LN, variations in the copy number of *FCGR3A* (genes associated with activating FC γ receptors) rather than *FCGR2B* (associated with inhibition of FC γ receptors) were associated with increased disease risk in anti-GBM patients [40]. More copies of FCGR3A may present phenotypically as a dosage effect to accelerate the development of anti-GBM disease [145,146]

### 2.5. Membranous Nephropathy

MN is characterised by the deposition of immune complexes within the kidney glomeruli. MN falls into either the primary or secondary category. Primary MN is characterised by the presence of phospholipase A2 receptor antibodies (anti-PLAR2, specific to the anti-phospholipase A2 receptor on podocytes) in 80% of cases, or, to a smaller extent, the development of antibodies against anti-thrombospondin type 1 domain containing 7A (anti-THSD7A), also located on podocytes (2–3% of cases). The development of secondary MN results from other diseases such as LN or cancer (commonly lung or colon), or as a result of therapeutic treatment (generally anti-inflammatory treatment such as ibuprofen or diclofenac) [147].

Clinically, MN commonly causes nephritic syndrome, which results in considerable proteinuria. It does not discriminate between geographical locations or ethnicities; however, males are more predisposed in a 2:1 ratio. MN is not categorised as a genetically inherited disease, but a growing body of evidence points towards a strong genetic association—in particular, HLA class II molecules encoded by alleles of the HLA-D locus on chromosome 6. Within Caucasian populations, there is a strong association with HLA-DR3 and HLA-DQA1, as well as *PLA2R1* on chromosome 2. If individuals are homozygous for these alleles, they are eighty times more likely to develop MN compared to individuals without the alleles [148]. In a Chinese cohort of patients, similar associations with HLA-DQA1 and *PLA2R1* were observed, where the likelihood of developing MN was eleven times greater than in those without the alleles [149]. Of critical importance is a landmark GWAS study conducted by Xie et al. of East Asian and European ethnicities [150]. This study demonstrated that the non-HLA genes, *NFKB1* and *IRF4*, together with HLA-associated DRB*1501 (for East Asians), DQA*0501 (in Europeans), and DRB1*0301 (both ancestries), could explain 32% of the risk of developing MN in East Asian populations and 25% of the risk in European populations. The identification of both *NFKB1* and *IRF4*, genes encoding transcription factors that activate downstream cascades that release proinflammatory cytokines, may provide new pathways to target therapeutically in the treatment of MN.

### 2.6. Minimal Change Disease

Minimal change disease (MCD) is one of the most common causes of nephrotic syndrome. It is characterised by significant proteinuria, with no evidence of glomerular histological damage or immune infiltrates when examined by light microscope. When examined by electron microscopy, podocyte foot process effacement is present. MCD is the most prevalent cause of nephrotic syndrome in children (10–50 cases per 100,000) and affects males more than females (2:1), this ratio disappears in adults and is closer to a 1:1 ratio [151]. A possible genetic cause for the disease has been proposed due to parent–child aggregation or between siblings [152]; however, these reports are rare. As podocytes have clear pathological damage, most of the research on MCD has centred around podocytes and proteins that are associated with podocyte biology. Large bodies of work have shown that nephrin, a key protein that is essential for maintaining the glomerular filtration barrier, is a likely candidate for the resulting pathology. Examination of kidney biopsies has demonstrated that nephrin loss is present in 34% of the samples, and that nephrin loss is associated with a lesser likelihood of remission (61%) [153]. Mutations in the nephrin-encoding gene *NPHS1* have been identified, but inconsistencies exist in reports on whether mutations in this gene result in the downregulation of nephrin when compared to control tissue [153,154]. *NPHS2* is the gene that encodes for the podocyte protein podocin and has been associated with the resulting podocyte pathology affecting the renal filtration barrier as well, but less is known about *NPHS2* than *NPHS1*. MCD can result in nephrotic syndrome in children, where four genes (*NHSP1*, *NHSP2, WT1* and *LAMB2*) are responsible for over 75% of the pathology. All of these genes are involved in the anatomical functioning of the podocyte. Of interest, those children that present with nephrotic syndrome in the first year of life all have mutations in these genes [155]. One of the limiting factors in examining genetic associations with MCD is the overlap between MCD and focal segmental glomerulosclerosis (FSGS); the overlapping of some of these genes has recently been reviewed by Bertelli et al. and will not be covered in this review [156]. Immune cells have been implicated to play a role in the pathology of MCD, with T cells, B cells, and dysfunctional T regulatory cells being identified in both human and mouse models of the disease [156]. However, a genetic association with these observations has not yet been explicated.

### 2.7. Membranoproliferative Glomerulonephritis

Membranoproliferative glomerulonephritis (MGPN) is more a descriptive term to describe a pattern of injury observed via electron and light microscopy. It can arise as a result of different causes and be referred to as primary or secondary (due to infections). Historically, MGPN was divided into different classes based on the histopathology. MGPN type I and III had very little C3 deposition or irregular patterns of Ig deposition (e.g., immune complexes), whereas MGPN type II was characterised by having predominately or exclusively C3 deposits. MGPN type II also had considerable electron-dense deposits, known as dense deposit disease (DDD), linked to defects in the alternative complement pathways; this is now termed C3 glomerulonephritis (C3Gs). For the purpose of this review, we will be discussing the known genetic associations of this subtype.

#### C3 Glomerulonephritis

C3Gs occurs at an incidence of 1–3 patients/million and, as such, is a rare disease and, although present in the older population, it is more routinely observed in childhood and adolescence [157]. As the name indicates, the disease is named to describe the defects arising in the complement system. The complement system itself comprises three different pathways that provide innate immunity to the host. The pathways are the classical pathway, the lectin pathway, and the alternative pathway. Although the pathways are different, they all result in the activation of the membrane attack complex, which punches holes in the plasma membranes of pathogens, causing destruction of the pathogen via osmolysis [158]. In C3Gs, it is the alternative pathway that is affected by hyperactivation, resulting in C3 deposits in the glomeruli and the fluid phase. The clinical manifestation of the disease is characterised by proteinuria and haematuria accompanied by kidney dysfunction. The prevalent genetic mutation in C3Gs is either a homozygous or heterozygous complement factor H (*CFH)* mutation. Over fifty *CFH* mutations have been described in these patients and involve frameshift mutations and nonsense and missense mutations [159]. C3, factor B (C*FB*), factor H (C*FH*), factor I (*CFI*), and FH-related (*CFHR)* have all been implicated. A mutation in any of them can arise in either the overexpression of C3 (*CFB*), a reduction in levels (*CFI*), or excess complement in sera (*CFH*) [158].

## 3. Genes Involved in Inefficient NET Clearance and Production in Autoimmune Kidney Diseases

The development of autoimmune kidney diseases is closely associated with the immune system, where a loss of immune homeostasis can have either direct or indirect renal effects, often resulting in ESKD [160,161,162]. When antigens within the kidneys are targeted, as in the case of anti-GBM disease and MN, they are referred to as direct immune-mediated renal disease [161]. In the context of infection or sterile autoimmunity, neutrophils are the first effector cells to be recruited into the kidneys, subsequently enabling the production and secretion of NETs, a unique form of neutrophil cell death [160,161]. However, in the context of autoimmunity, the production of NETs is injurious.

NETs are a type of neutrophil-mediated cell death, where web-like structures composed of decondensed DNA together with histones and neutrophil granule proteins (e.g., MPO, neutrophil elastase (NE); lactoferrin and metalloproteinases (MMPs)) are released extracellularly [5,163,164,165]. It is a caspase-independent process that causes the neutrophil lobules to lose their shape and breaks down their nuclear envelopes, facilitating the formation of multiple vesicles and the release of nuclear and mitochondrial DNA [5,163,164,165,166,167]. Pathogen-associated molecular patterns (PAMPs) derived from various microbial species have been found to potently induce NET production [165,166]. Thus, during infections, NETs are released to neutralise, immobilise, and kill foreign pathogens [165,166,167,168]. Furthermore, during NETosis, neutrophil granule proteins remain enzymatically active. For example, NE can cleave microbial virulence factors, while MPO and DNA derived from NETs have been found to be vital for eliminating *S. aureus* and *Pseudomonas aeruginosa*, respectively [167,168,169,170]. Although the secretion of granular proteins at a high local concentration can be protective due to the amplification of the antimicrobial effects of neutrophils, it can also serve as a double-edged sword, causing serious inflammation and the destruction of bystander tissues [5,163,168]. 

It has been proposed that the dysregulation of apoptosis or the clearance of apoptotic material serves as a source of externalised autoantigens [171]. The findings of Hakkim et al. allude to the idea that NETs themselves are a source of autoantigens and are likely to be more immunogenic than apoptotic material [172]. It is believed to be more immunogenic as the process of NETosis increases the proximity of genomic DNA to reactive oxygen species (ROS), resulting in the oxidation of nucleic acids that are more resistant to nuclease-mediated degradation [171,172]. Extracellular DNA derived from NETs serves as a danger-associated molecular pattern (DAMP) that activates and induces inflammation through the activation of TLR9, cyclic-di-GMP-AMP (cGAMP) synthase (cGAS)–STING, and the inflammasome [5,173,174,175]. Additionally, through NETosis, NE and MPO are deposited extracellularly [5,163,176]. NE can induce tissue remodelling through the degradation of host extracellular matrix proteins. Under normal circumstances, NE is regulated through endogenous inhibitors; however, the release of MMPs during inflammation degrades and inactivates NE inhibitors [176]. Hakkim et al. have identified a correlation between the development and exacerbation of SLE and an impairment in NET degradation, whereby neutrophils derived from healthy participants were found to have complete degradation of NETs, while SLE patients were found to have higher antibody titres bound to NETs or were found to have inhibitors that prevented NET degradation [172]. Furthermore, NET-derived MPO is biologically active and harder to degrade, leading to its accumulation within the kidneys and causing direct glomerular endothelial tissue injury [5,163].

Assessing genetic variants associated with NETs may help to identify potential molecular mechanisms in NET formation and the subsequent downstream effects that result in kidney autoimmune pathogenesis. GWAS analysis identified SNPs annotated to four genes (Kinesin Family Member 26B (*KIF26B****)****, CDK19, CATSPERB,* and *AC027119.1*) that were suggested to be associated with plasma MPO-DNA complexes, which are biomarkers for NETs. The suggestive SNP located on the *KIF26B* (Kinesin Family Member 26B) gene is of particular importance given that its activation has been proposed to activate the phosphatidylinositol 3-kinase (PI3K) and Akt (PI3K/Akt) signalling pathways (summarised in Figure 2) [177]. The PI3K/Akt signalling pathway can activate the enzyme NADPH oxidase, an essential enzyme in NETosis, facilitating the translocation and activation of granular proteins (MPO and NE), as well as increasing the expression of Peptidyl Arginine Deiminases (PADs) [178]. Through exome sequencing, it was found that a rare variant in the Transmembrane Serine Protease 13 (*TMPRSS13****)*** gene was correlated to significant levels of MPO-DNA complexes. This is understandable given that this gene plays a critical role in activating the membrane fusion activity of influenza, particularly the glycoprotein hemagglutinin (HA). Gene-based analysis revealed 10 additional genes (Olfactory Receptor Family 10 Subfamily H Member 1 (*OR10H1), RP11-461L13.5, RP11-24B19.4, RP11-461L13.3, KHDRBS1, ZNF200, RP11-395I6.1, RP11-696P8.2, RGPD1,* and *AC007036.5*) that were involved, with the strongest correlation for MPO-DNA complex levels observed in the *OR10H1* (Olfactory Receptor Family 10 Subfamily H Member 1) gene. Interestingly, eight of the identified genes (*RGDP1, KIF26B, KHDRBS1, ZNF200, CATSPERB, CDK19, OR10H1,* and *TMPRSS13)* were found to be interrelated, exhibiting gene–gene functional interactions that were involved in inflammatory processes [177]. 

*TNFAIP3* encodes A20, a negative regulator of NF-κB-induced signalling, whereby the non-synonymous polymorphism rs2230926 in the A20 de-ubiquitinase (DUB) domain precipitates the development of autoimmune diseases including SLE [182,183,187]. Odqvist et al. have since discovered that genetic variants in the A20 DUB domain upregulate the expression of the gene *PADI*4, responsible for encoding Peptidyl Arginine Deiminase-4 (PAD4) [187]. PAD4 is essential for the citrullination of histones and the decondensation of chromatin [5,163,184], which facilitates the formation and secretion of NETs. Whilst NETosis can be regulated through PAD4 inhibition, targeting *TNFAIP3* specifically through gene therapy could prove to be more beneficial, especially considering that both NF-κB signalling and PAD4 activity can be simultaneously inhibited, allowing for the attenuation of inflammation and NETosis. 

α-1 antitrypsin (AAT) is a serine protease inhibitor produced in hepatocytes. AAT is the most prominent protease inhibitor found in plasma. AAT is encoded by the *SERPINA1* gene. AAT inhibits multiple serine proteases but, in particular, neutrophil elastase. Therefore, any mutations in the *SERPINA1* gene can result in excessive neutrophil elastase (NE) deposition [182]. This has critical implications for injury to blood vessels and epithelial cells, which are directly affected with the release of neutrophil NE. As NE is also critical in the production of NETs, a deficiency in AAT can cause uncontrolled NET release. AAT has been found to be deficient in patients with AAV and is correlated with increased NET formation in type 1 diabetes [188,189].

The removal of NETs is crucial in maintaining homeostasis and preventing the development of kidney autoimmune diseases. DNases are fundamental proteins involved in this process. The DNase complex is composed of three enzymes, DNase I, DNase II, and DNase 1L3, and is responsible for digesting and removing circulating DNA [185]. Mutations in the particular DNase genes that affect the activity of the subsequent DNAse released (e.g., SNPs indicated in Table 1 and Table 2) have been linked to immunological syndromes. Ineffective production or reduced production of DNase I leads to the accumulation of NETs and extracellular DNA that can trigger autoimmune GN [101,185,186]. This is due to the proinflammatory nature of extracellular DNA release, activating PRRs and initiating proinflammatory cell signalling pathways (e.g., NFқB). TLR9 in particular recognises dsDNA as a DAMP and has been implicated in aggravating GN in animal models and is expressed significantly in biopsies of patients with MPO-ANCA GN [190,191,192]. Kenny et al. previously demonstrated that DNase I knockout mice spontaneously developed lupus-like disease, with mice exhibiting elevated levels of autoantibodies and kidney damage within 12 months, while the development of LN has been associated with mutations in the DNase I and DNase 1L3 genes, subsequently affecting the clearance of NETs and leading to their accumulation [185,186,193]. Fenton et al. also demonstrated that DNase activity was significantly downregulated in murine kidneys as nephritis exacerbated [194,195,196]. Clinical trials using recombinant DNase have been demonstrated to be safe and efficacious for severely ill cystic fibrosis patients [197]. Thus, targeting the DNase complex as a prophylactic or early interventional gene therapy **for kidney autoimmunity may prove to** be highly beneficial.

## 4. Conclusions

Currently, autoimmune kidney diseases have suboptimal treatment, which contributes significantly to morbidity and mortality. Although each of the diseases discussed here has different mechanisms of pathogenesis and genetic associations, shown by GWAS studies, the current treatment remains unspecific and does not address the primary underlying pathology of each disease. Treatment consists mainly of induction therapy using cyclophosphamide, followed by high-dose glucocorticoids, which renders patients significantly immunocompromised and increases their risk of life-threatening infections, with the potential of developing malignancies. The studies discussed in this review may provide insights and create a foundation for basic science to explore the association between genetic risk factors and their resulting pathologies, to inform the basis for novel and targeted treatments of autoimmune kidney diseases. 

## Figures and Tables

**Figure 2 genes-14-01028-f002:**
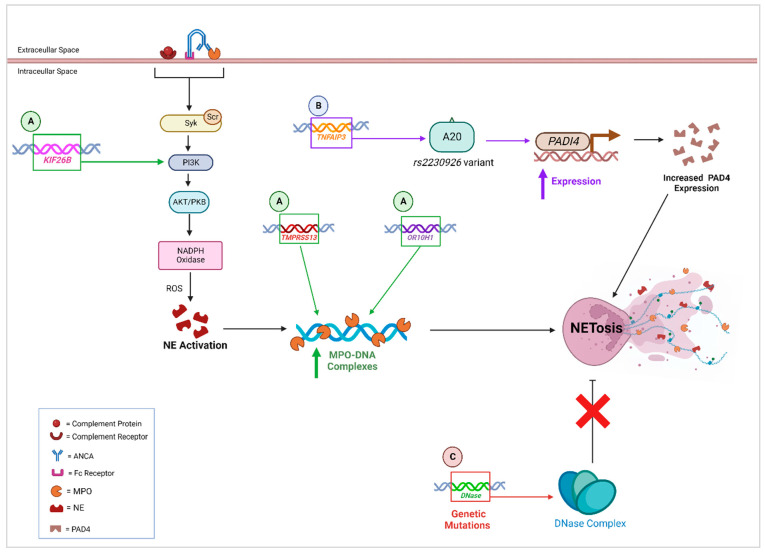
Overview of SNPs affecting NET formation. (**A**) *KIF26B* activates PIK3/Akt signalling pathway, resulting in the activation of NADPH oxidase, which, together with the influence of reactive oxygen species (ROS), induces neutrophil elastase (NE) activation. This results in the upregulation of MPO-DNA complexes and an increase in NETosis [177,178,179,180]. The genes *TMPRSS13* and *OR10H1* have also been found to increase the expression of MPO-DNA complexes [177]. (**B**) *TNFAIP3* encodes the protein A20. Genetic variants in the A20 de-ubiquitinase (DUB) domain (e.g., *rs2230926*) result in the increased activation of *PADI4,* facilitating an increase in Peptidyl Arginine Deiminase-4 (PAD4) and NETosis [5,163,181,182,183,184]. (**C**) The DNase complex is a negative regulator of NETosis; however, genetic mutations in the *DNase* genes will cause the accumulation of NETs [185,186]. Figure created using BioRender.com (15 November 2022).

## Data Availability

Not applicable.

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
