# Peer review of "Deciphering the Genetic Code of Autoimmune Kidney Diseases"

_genes, 2023, doi:10.3390/genes14051028_

Round 1

Reviewer 1 Report

 Authors have reviewed genetic associations of autoimmune kidney diseases- Lupus Nephritis, anti-neutrophil cytoplasmic associated vasculitis (AAV), anti-glomerular basement disease IgA nephropathy, and membranous nephritis. These genetic associations with increased risk of disease including SNP’s  in HLA and genes involved in inflammation.

1. What is the main question addressed by the research?
Recently characterized  single nucleotide polymorphisms (SNPs) in the pathogenesis of  autoimmune kidney disease

2. Do you consider the topic original or relevant in the field? Does it
address a specific gap in the field?
Yes, it is relevant in the field. Association of single nucleotide polymorphisms and signaling mechanisms in the pathogenesis of autoimmune kidney disease.

3. What does it add to the subject area compared with other published
Material? Recently characterized single nucleotide polymorphisms (SNPs) which influence autoimmune kidney disease pathogenesis neutrophil extracellular traps that induce glomerular nephritis.

4. What specific improvements should the authors consider regarding the
methodology? What further controls should be considered?
None

5. Are the conclusions consistent with the evidence and arguments presented
and do they address the main question posed?
Yes

6. Are the references appropriate?
Yes

7. Please include any additional comments on the tables and figures.
Title or description legend  for the figure.2 is missing, authors consider including that in the final version.

Comments:

1. Jacob M, 2018 et.al (https://doi.org/10.1159/000488069), “CTLA4 -318/C/T SNP was associated with an increased risk to develop IgAN, while the CT60 G/A genotype significantly associated with the risk for higher proteinuria suggesting a possible role for CTLA-4 in IgAN”., Authors consider to include this report

Author Response

Thank you for your insightful comments.

We have now amended the manuscript to add a description of table 2 as per your suggestion.

Thankyou for suggesting including the SNP for CTLA, we have now included this reference and discussed on page 9.

Reviewer 2 Report

1 The title covers too much and is overly descriptive.

2 Authors did not carefully check the details of the review. There are many typos, such as Line 65 affect.

3 Abbreviations should follow the full name of a concept not the opposite such as IL10 (interleukin 10).

4 The same abbreviations such as LN ,AAV appeared many times in the review. Abbreviations should be used at the first time not afterwards such as T helper 1 (Th1) cells, neutrophil extracellular traps (NETs)….

5 Line 184-187 was redundant.

6 Strong associations were shown for both MPO-AAV and PR3- 230 AAV with FCGR3B copy number, where differences in the copy numbers were associated 231 Disease: What did the copy number mean?

7 Line 264, no reference.

8 I do not think the part ‘Neutrophil Extracellular Traps and Gene Associations’ were closely associated with autoimmune kidney diseases.

9 A table should be added with SNPs, protein ids, and main functions.

Author Response

The title covers too much and is overly descriptive.

Thank you for this comment. We have tried to remove words from the title but when we did it lost its meaning. If we remove any further words from this title readers will not know what the article is about (8 word title is within title limits, and is shorter than the average titles for review articles).

2 Authors did not carefully check the details of the review. There are many typos, such as Line 65 affect.

This was an oversight on our part. Thank you for your suggestions, a careful review of the manuscript has been conducted and typos corrected.

3 Abbreviations should follow the full name of a concept not the opposite such as IL10 (interleukin 10).

Thank you for your suggestions, a careful review of the manuscript has been conducted and this has been corrected.

4 The same abbreviations such as LN ,AAV appeared many times in the review. Abbreviations should be used at the first time not afterwards such as T helper 1 (Th1) cells, neutrophil extracellular traps (NETs)….

Thank you for your suggestions, a careful review of the manuscript has been conducted and this has been corrected.

5 Line 184-187 was redundant.

Thank you for your suggestion this has been removed.

6 Strong associations were shown for both MPO-AAV and PR3- 230 AAV with FCGR3B copy number, where differences in the copy numbers were associated 231 Disease: What did the copy number mean?

We have amended the manuscript to be clearer. This is now evident on page 8. We have changed the section to read as follows: Low copy number of FCGR2B was found to be associated with systemic inflammation, which commonly affected the glomerulus [39]. As FCGR3B is responsible for the tethering of neutrophils and eosinophils to immune complexes and subsequent clearance [69]. Individuals with low copy numbers of the gene may have reduced clearance of immune complexes, leading to their deposition in the glomerular microvasculature, triggering an inflammatory response, which, if chronic, may develop into autoimmunity. Moreover, enhanced neutrophil activation by ANCA may be associated with reduced or absent expression of FCGR3B [39], providing further evidence for the involvement of altered copy number in the pathogenesis of ANCA vasculitis.

7 Line 264, no reference.

A reference has now been included.

8 I do not think the part ‘Neutrophil Extracellular Traps and Gene Associations’ were closely associated with autoimmune kidney diseases.

Thank you for this comment. We have amended the title to be more descriptive of the following content it now reads as :

3. Genes involved in inefficient NET clearance and production in autoimmune kidney diseases

Further it has been widely published that the failure to remove NETs in autoimmune kidney diseases contributes to inflammation and the deposition of over 70 known autoantigens. Genes control enzymes that are critical in NET induction , and for the production of DnaseI to clear NETs. Our whole research program is on the role of neutrophil extracellular traps in autoimmune kidney diseases- all my publications related to this topic can be read here: https://scholar.google.com/citations?user=jGx3vREAAAAJ&hl=en

The role of NETs in autoimmune kidney diseases has been further reviewed in the publication here: Neutrophil extracellular traps: A potential therapeutic target in MPO-ANCA associated vasculitis? KM O'Sullivan, SR Holdsworth. Frontiers in Immunology 12, 635188

9 A table should be added with SNPs, protein ids, and main functions.

Thank you this is a great suggestion and has been included and is now the new Table 2.

Reviewer 3 Report

The manuscript entitled “Deciphering the genetic code of autoimmune kidney diseases” by Huang et al. summarizes the most recent genetic associations of five major autoimmune kidney diseases (LN, AAV, IgA Nephropathy, Anti-GBM, and Membranous Nephropathy). In addition, they review the function of the SNPs of NETs-related genes in LN, AAV, and Anti-GBM. This well-organized review publication offers an in-depth understanding of the genetic relationship in autoimmune kidney disorders. However, there are a few issues that need to be addressed by the authors: 

1.     The data sources in Table I need to be marked with references. Please carefully review Table 1 as some of the genes referenced in the paper, such as DNASE1L3 and IL23R, are not included. Besides, it should be CTLA4, not CTLA44.

2.     Different genes of HLA are associated with different autoimmune diseases (lines 199-207). It is recommended to provide a deeper insight into the molecular mechanisms that may contribute to this distinction in disease pathogenesis.

3.     As population information is crucial for the GWAS investigation, it would be better to include the population information of the analysis in the review (such as line 263).

4.     This paper contains some erroneous abbreviations. Only use abbreviations for phrases used three or more times in the manuscript (such as PBMC in line 147 is unnecessary). Also, Abbreviations should be avoided in the abstract unless a term is used multiple times. Furthermore, define abbreviations at first mention (such as line149 Th1)

5.     A subtitle in lines 198 and 208 is recommended.

6.     It will be more reader-friendly to explain the function of PI3K/Akt in NET formation (line 364)

7.     As NETs are also linked to other diseases, thus the title of 3 should be more specific to the topic on which this manuscript focuses.

8.     Line 335 includes two periods.

9.     Genetic variants contain pathogenic variants and benign variants. It should be more cautious about drawing the conclusion in line 424.  

Author Response

  1. The data sources in Table I need to be marked with references. Please carefully review Table 1 as some of the genes referenced in the paper, such as DNASE1L3 and IL23R, are not included. Besides, it should be CTLA4, not CTLA44.

Thank you for these suggestions. References have been included to the table, as well as an additional table highlighting the functional relevance of each gene.

  1. Different genes of HLA are associated with different autoimmune diseases (lines 199-207). It is recommended to provide a deeper insight into the molecular mechanisms that may contribute to this distinction in disease pathogenesis.

Thank you this suggestion. We have made amendments to this paragraph. As per below:

HLA-DPB genes belong to the HLA class II beta chain paralogues, where the class II molecule consists of an alpha (DPA) and beta (DPB) chains that form a heterodimer, this complex is expressed on antigen presenting cells and functions to present peptides. Polymorphisms can occur within the alpha or beta chain, which can alter the peptide binding specificity, which is associated with a higher risk of GPA. Like HLA-DPB, HLA-DQB genes, which were found to be associated to MPA, also belong to the HLA class II beta chain paralogues, these heterodimers consist of DQA alpha and DQB beta chains instead of DPA and DPB chains, and polymorphisms within these chains also affect binding affinity [66]. Differences in the amino acid sequences of HLA proteins may affect the way they bind and present peptides, potentially leading to differences in the types of immune responses they stimulate. HLA alleles can bind a wide range of peptides, and different alleles may have different peptide-binding preferences. Studies showed that these polymorphisms had strong associations were with ANCA specificity, with others showing associations with disease severity, prognosis and relapse [127, 128]

  1. As population information is crucial for the GWAS investigation, it would be better to include the population information of the analysis in the review (such as line 263).

This was an oversight on our part. This has been corrected and the population information has been added to the manuscript as follows:

IL23R and IL17A were recently associated with IgAN susceptibility for the first time in a Chinese Han population [58]. A comparative analysis between genotype distributions, clinical indexes and pathological grades was carried out between 164 IgAN patients and 192 healthy controls. The analysis revealed that the GG genotype in rs7517847 was associated with a lowered IgAN risk compared to the TT genotype. Another polymorphism in rs2275913 was detected, where the AA genotype was related to a decreased IgAN risk compared to the GG genotype. IL23R subunits were shown to be present on γδ T cells and Th17 cells, which produce IL17A [143]. γδ T cells have been found to infiltrate the kidney in patients with progressive IgAN [103] and the imbalance of Tregs and Th17 cells, was also found in IgAN patients [104].

  1. This paper contains some erroneous abbreviations. Only use abbreviations for phrases used three or more times in the manuscript (such as PBMC in line 147 is unnecessary). Also, Abbreviations should be avoided in the abstract unless a term is used multiple times. Furthermore, define abbreviations at first mention (such as line149 Th1). 

Thank you for this suggestion. These erroneous abbreviations have been reduced or removed throughout the manuscript.

  1. A subtitle in lines 198 and 208 is recommended.

Thank you these have been amended to  numbered subtitles. They now read as follows;

2.2.1 HLA region SNPs

2.2.2 Non-HLA region SNPs

  1. It will be more reader-friendly to explain the function of PI3K/Akt in NET formation (line 364)

Thank you for this suggestion we have made a reader friendly explanation of the role of this pathway in NET formation as below.

The suggestive SNP located on the KIF26B (Kinesin Family Member 26B) gene is of particular importance given that its activation has been proposed to activate phosphatidylinositol 3-kinase (PI3K) and Akt (PI3K/Akt) signalling pathways (summarised in Figure 3)[177] The PI3K/Akt signalling pathway can activate the enzyme NADPH oxidase, an essential enzyme in NETosis facilitating the translocation and activation of granular proteins (MPO and NE), as well as increasing the expression of Peptidyl Arginine Deiminases (PADs) [178]

  1. As NETs are also linked to other diseases, thus the title of 3 should be more specific to the topic on which this manuscript focuses.

Thank you for this suggestion. We agree this makes the title more specific. WE have changed the title to:

Genes involved in inefficient NET clearance and production in autoimmune kidney diseases

  1. Line 335 includes two periods.

Thank you, the spare period has been removed.

  1. Genetic variants contain pathogenic variants and benign variants. It should be more cautious about drawing the conclusion in line 424. 

Thank you for this suggestion. We have made this paragraph more specific to genes mutations that decrease the activity of DNase I and given specific relevant experiments that support this.

Mutations in the particular DNase genes that affect the activity of the subsequent DNAse released (eg SNPs indicated in table 1 and 2) have been linked to immunological syndromes. Ineffective production or reduced production of DNase I leads to the accumulation of NETs and extracellular DNA that can trigger autoimmune GN [98, 185, 186]. This is due to the proinflammatory nature of extracellular DNA release activating PRRs, and initiating proinflammatory cell signalling pathways. TLR9 in particular recognises dsDNA as a DAMP and has been implicated in aggravating GN in animal models and is expressed significantly in biopsies of patients with MPO-ANCA GN [191-193].

Round 2

Reviewer 2 Report

Immunoglobulin- and complement-mediated glomerular diseases

with a membranoproliferative glomerulonephritis (MPGN) pattern

of injury are known to be accompanied by autoimmune dysregulation. Besides, as an autoimmune kidney disease linked to T cell dysregulation, MCD is widely recognized. Do MPGN and MCD exhibit any genetic advancement?

For example:

C3 glomerulonephritis and C3 DDD:Mutations in complement regulatory proteins: CFH, CFI, CFHR5. Mutations in complement factors: C3. Antibodies to complement factors: C3, C4, and C5 nephritic factors. Antibodies to complement regulatory proteins: CFH, CFl, CFB

Author Response

Immunoglobulin- and complement-mediated glomerular diseases with a membranoproliferative glomerulonephritis (MPGN) pattern of injury are known to be accompanied by autoimmune dysregulation. Besides, as an autoimmune kidney disease linked to T cell dysregulation, MCD is widely recognized. Do MPGN and MCD exhibit any genetic advancement?For example:

C3 glomerulonephritis and C3 DDD:Mutations in complement regulatory proteins: CFH, CFI, CFHR5. Mutations in complement factors: C3. Antibodies to complement factors: C3, C4, and C5 nephritic factors. Antibodies to complement regulatory proteins: CFH, CFl, CFB. 

Thank you for these suggestions. We have made modifications to the manuscript and added complement mediated glomerulonephritis and MCD. Please note that there has been no new advancement or identification of SNPs ( just a few case studies) the focus of this manuscript, so the genes have not been added to table 1, but been included  in table 2 to explain what the function of the gene is.

You can find these new changes both in the track changed word document and the clean PDF document on lines 354-413.

You have noted that moderate English changes are required. Could you please indicate where this is required. All 3 authors first language is English (British English, not American English)- so we will need guidance on where you think the English is incorrect or needs improvement-is it an American versus British English that is the issue? If you could list the examples we will amend them accordingly.